Utilizing network pharmacology and experimental validation to investigate the underlying mechanism of phellodendrine on inflammation

Hu Lili 1 hulili0204@163.com
Wang Jue 1
Wu Na 1
Zhao Xiaoge 2
Cai Donghui 1
1 Shanxi University of Chinese Medicine , Jinzhong , China
2 Xi’an Jiaotong University , Xi’an , China
Sistla Srinivas
Electronic publication date: 2022 Sep 23
Publication date: 2022
Volume: 10
Electronic Location ID: e13852
Received 2022 Mar 31; Accepted 2022 Jul 16
Copyright: © 2022 Hu et al.
Copyright year: 2022
Copyright holder: Hu et al.
License: This is an open access article distributed under the terms of the Creative Commons Attribution License, which permits unrestricted use, distribution, reproduction and adaptation in any medium and for any purpose provided that it is properly attributed. For attribution, the original author(s), title, publication source (PeerJ) and either DOI or URL of the article must be cited.
License URL: https://creativecommons.org/licenses/by/4.0/

Keywords: Phellodendrine, Inflammation, Network pharmacology, Multiple targets, Mechanism

Funding: 2020 Science and Technology Innovation Project of Shanxi Provincial Department of Education 2020L0420 Shanxi University of Chinese Medicine 2020PY-JC-06, 2021PY-JC-06 This work was financially supported by the 2020 Science and Technology Innovation Project of Shanxi Provincial Department of Education (No. 2020L0420) and the Cultivation Project in Scientific and Technological Innovation of Shanxi University of Chinese Medicine (No. 2020PY-JC-06, 2021PY-JC-06). The funders had no role in study design, data collection and analysis, decision to publish, or preparation of the manuscript.

==============================
Background

Phellodendrine, one of the characteristic and important active components of Cortex phellodendri, has been proven to show anti-inflammatory effects. However, the underlying mechanism of phellodendrine on inflammation remains largely unclear.

Aim of the study

In this study, network pharmacology and experimental validation were used to explore the underlying mechanism of phellodendrine on inflammation.

Materials and Methods

PubChem and SwissADME database were used to evaluate the drug-likeness and other characteristics of phellodendrine. The targets of phellodendrine for the treatment of inflammation were analyzed with multiple databases. Other extensive analyses including protein–protein interaction, Gene Ontology, and Kyoto Encyclopedia of Genes and Genomes pathway enrichment were accomplished with the STRING database, Cytoscape software, and DAVID database. Moreover, the effect of phellodendrine on anti-inflammation was proven in RAW264.7.

Results

The network pharmacology results indicated that phellodendrine had drug potential. Phellodendrine acted directly on 12 targets, including PTGS1, PTGS2, HTR1A, and PIK3CA, and then regulated cAMP, estrogen, TNF, serotonergic synapse, and other signaling pathways to exert anti-inflammatory effects. The experimental results showed that phellodendrine reduced the levels of IL-6 compared with the LPS group in 24 h and changed the mRNA expression of PTGS1, PTGS2, HSP90ab1, AKT1, HTR1A, PI3CA, and F10.

Conclusion

Our research preliminarily uncovered the therapeutic mechanisms of phellodendrine on inflammation with multiple targets and pathways. Phellodendrine may be a potential treatment for inflammation-related diseases related to the cAMP and TNF signaling pathways.

Introduction

Inflammation is a tightly regulated system of markers that can orchestrate a response to injury or infection and promote healing (Szabo, Burns & Lantrip, 2022). Inflammation is implicated in the pathogenesis of many chronic physical and mental conditions, including cancer, coronary heart disease, diabetes mellitus, obesity, ulcerative colitis, rumination, depression, anxiety, borderline personality disorder, and schizophrenia (Szabo, Burns & Lantrip, 2022; Liu et al., 2021; Chu et al., 2021; Berk et al., 2013; Su et al., 2021). Many studies have reported the treatment of various diseases via anti-inflammatory methods. As early as the nineteenth century, studies reported clinical evidence of the advantageous effect of anti-inflammatory treatment in diabetes (Ebstein, 1876). IL-1 inhibitors (anakinra), NF-κB inhibitors (salsalate), and TNF-a (etanercept) were used to ameliorate insulin resistance (Rohm et al., 2022). The majority of the literature indicated that anti-inflammatory agents (non-steroidal anti-inflammatory drugs and cytokine inhibitors) show an antidepressant effect in patients with major depressive disorder (Bai et al., 2020).

Phellodendrine, which was discovered by a Japanese scholar in 1962, is one of the characteristic and important active components of Cortex phellodendri (Li et al., 2016). Cortex phellodendri (Huangbo in Chinese medicine) derived from the dried bark of Phellodendron chinensis Schneid. or Phellodendron amurense Rupr. (Family Rutaceae) (Chen et al., 2010) is a traditional herbal medicine widely used in China for over thousands of years. Traditionally, Cortex phellodendri is used to reduce fever, eliminate steam, and remove toxins (Sun, Lenon & Yang, 2019). Cortex phellodendri shows therapeutic effects in multiple diseases such as meningitis, cirrhosis, dysentery, pneumonia, and tuberculosis (Jiang et al., 2016; Sun, Lenon & Yang, 2019). Recent pharmacological studies showed that Cortex phellodendri has antibacterial, antidiarrheal, anti-apoptotic, and anti-inflammatory effects (Chen et al., 2010; Jung et al., 2009; Bae, Shim & Kim, 2011). An increasing number of researchers are focusing their attention on the effect of phellodendrine on inflammation and inflammation-related diseases. For example, phellodendrine inhibits the cellular immune response as an immunosuppressor (Hattori et al., 1992; Mori et al., 1994). Phellodendrine shows protective activity against AAPH-induced oxidative stress, which is similar to the inflammatory response in vivo (Li et al., 2016). Previous network pharmacology research has demonstrated that phellodendrine can be used against diabetes mellitus through the calcium signaling pathway, cGMP-PKG signaling pathway, and cAMP signaling pathway (Zhang et al., 2021). Phellodendrine may inhibit the proliferation or the migration of macrophages and cytotoxic T lymphocytes in the glomeruli, showing high effectivity in anti-GBM nephritis (Hattori et al., 1992). Phellodendrine can reduce the intestinal damage of ulcerative colitis, which is a gastrointestinal inflammatory disease (Su et al., 2021). However, the underlying comprehensive mechanism of phellodendrine on the common status of inflammation is unclear.

In the present study, we aimed to elucidate the mechanism of phellodendrine on inflammation through network pharmacology and experimental validation (Fig. 1). We evaluated the druglikeness and other characteristics of phellodendrine with the SwissADME database and predicted targets of phellodendrine against inflammation through multiple databases. A protein–protein interaction (PPI) network was constructed with the common targets. Gene ontology (GO) and Kyoto Encyclopedia of Genes and Genomes (KEGG) pathway enrichment were analyzed using the DAVID database. Phellodendrine’s anti-inflammatory effects were validated in RAW264.7. IL-1β and IL-6 were detected via ELISA. The expression levels of target genes were confirmed using quantitative real-time PCR. Results of this study may provide a possible therapeutic mechanism of phellodendrine on inflammation and inflammation-related diseases.

Figure 1 Graphical abstract.

The flow diagram of the research process. The whole research process includes two parts, network pharmacology and experimental validation.

Materials and Methods

Drug-likeness and other characteristics of phellodendrine

Lipinski’s rule of five (RO5) is used to evaluate the drug-likeness of oral drugs in humans. The parameters are mainly composed of molecular weight (MW, lower than 500 g/mol), topological polar surface area (TPSA, less than 140 A2), octanol–water partition coefficient (log P0/W (MLOGP), lower than 5), number of hydrogen-bond acceptors (nHAcc, less than 10), number of hydrogen-bond donors (nHDon, less than 5), and number of rotatable bonds (lower than 10). Other characteristics such as log S (ESOL), GI absorption, BBB permeant, log Kp (skin permeation), and bioavailability score were recorded. The chemical structure and canonical SMILES of phellodendrine were obtained from the PubChem database (https://pubchem.ncbi.nlm.nih.gov/). To estimate the drug-likeness properties and other characteristics of phellodendrine, we imported the Canonical SMILES

C[N+]12CCC3=CC(=C(C=C3C1CC4=CC(=C(C=C4C2)OC)O)O)OC of phellodendrine into the SwissADME database (http://www.swissadme.ch/) (Daina, Michielin & Zoete, 2017).

Collection of phellodendrine-related targets and inflammation-related targets

TCMSP (https://old.tcmsp-e.com/tcmsp.php) and SwissTargetPrediction (http://www.swisstargetprediction.ch/) were required to collect the phellodendrine-related targets depending on chemical names (phellodendrine) and canonical SMILES (C[N+]12CCC3=CC(=C(C=C3C1CC4=CC(=C(C=C4C2)OC)O)O)OC) (probability > 0.09724). The union of targets from the two databases was the phellodendrine-related targets.

Inflammation-related targets were selected from the OMIM (http://www.omim.org), Drugbank (https://go.drugbank.com/), and GeneCards (https://www.genecards.org/) databases by searching “inflammation” as a key word. The obtained protein target name was converted into gene name (office gene symbol) using the UniProt database (https://www.uniprot.org). Duplicate genes were deleted.

Both phellodendrine targets and inflammation targets were imported into Venny 2.1 software (https://bioinfogp.cnb.csic.es/tools/venny/index.html) to obtain the common targets of phellodendrine against inflammation.

Network construction of phellodendrine inflammation common targets

The PPI network was built using the STRING database (https://www.string-db.org/). Common targets were uploaded into the STRING database, Homo sapiens was selected in the term of organism, and only a medium confidence score of 0.400 was chosen. Common target interaction information obtained from the STRING database was integrated and visualized by Cytoscape (version 3.7.2) software.

GO and KEGG pathway enrichment analyses

In our study, GO and KEGG pathway enrichment analyses were carried out using the DAVID database (DAVID Bioinformatics Resources 6.8, https://david.ncifcrf.gov/), and statistical significance was set at P < 0.05. A heatmap was plotted by bioinformatics (http://www.bioinformatics.com.cn), which is an online platform for data analysis and visualization.

Validation of phellodendrine’s anti-inflammatory effect by preventing inflammation in RAW264.7

Cell culture

RAW264.7 cells were purchased from the Shanghai Institute of Cell Biology, Chinese Academy of Sciences (Shanghai, China). Cells were cultured in Dulbecco’s modified eagle’s medium (HyClone, New York, NY, USA) containing 10% fetal bovine serum (Gibco, New York, NY, USA) and incubated at 37 °C with 5% CO2.

Cell viability

MTT assay was used to measure cell viability. In brief, the RAW 264.7 cells were seeded into 96-well plates with a density of 5,000 per well and then incubated at 5% CO2 and 37 °C overnight. Various concentrations of phellodendrine (5, 10, 20, 40, 80, and 160 mg/L) were added to the cells in the presence or absence of LPS (1 mg/L) for 24, 48, and 72 h. The cells were subjected to MTT (20 μL, 5 mg/mL) and then incubated for 4 h. The supernatant was discarded completely, and the formed formazan was dissolved by adding 150 μL of DMSO. Finally, optical density (OD) was measured at 492 nm.

Cytokine measurement

To detect cytokine production, we seeded RAW 264.7 cells into 96-well plates (1 × 104 cells/well) overnight. The following groups and protocols were consistent with MTT assay. Cytokines in each group were detected by the IL-1β and IL-6 ELISA kits (Shanghai Jining Shiye Co., Ltd., Shanghai, China) in accordance with the manufacturer’s instructions.

Quantitative real-time PCR

Total RNA was extracted from each group and then reverse-transcribed into cDNA using PrimeScript™ RT reagent Kit (Takara Biomedical Technology, Beijing, China). The primers of the target genes and β-actin were used as shown in Table 1. Quantitative real-time PCR was performed with TB Green® Premix Ex Taq™ (Takara Biomedical Technology, Beijing, China). The relative expression levels were computed using the 2−ΔΔCT method.

Table 1 Summary of the qRT-PCR primer sequences.

Gene	Primers	Sequences	Product length	
Ptgs2	Forward	CATCCCCTTCCTGCGAAGTT	178	
	Reverse	CATGGGAGTTGGGCAGTCAT		
Adrb2	Forward	TGGTTGGGCTACGTCAACTC	152	
	Reverse	TCCGTTCTGCCGTTGCTATT		
Ptgs1	Forward	CCAGGAGCTCACAGGAGAGA	197	
	Reverse	ACTCTGGGGAACAGATGGGA		
F10	Forward	TCTACCAGCTGGGAAGCAG	131	
	Reverse	CCTTCTGATTGGAGCCCTGG		
Akt1	Forward	CCCATCTGAGTCCACAGCAA	81	
	Reverse	CCGCAGCGTCTGGACA		
HSP90ab1	Forward	AGATTCCACTAACCGACGCC	150	
	Reverse	TGCTCTTTGCTCTCACCAGT		
HTR1A	Forward	TACTCCACTTTCGGCGCTTT	182	
	Reverse	GGCTGACCATTCAGGCTCTT		
PIK3CA	Forward	AGAAGCCGGAGCGGCA	116	
	Reverse	GTTCACCCGAAGATGGTCGT		

Statistical analysis

Data in the cell experiments were presented as means ± SEM and analyzed using SPSS 17.0. Differences among groups were analyzed by one-way ANOVA and Tukey’s post-hoc test. P < 0.05 was considered statistically significant for analysis.

Results

Molecular properties of phellodendrine

As mentioned above in the Methods section, we examined 11 properties. The MW of phellodendrine was 342.41 g/mol, nHAcc was 4, nHDon was 2, logP0/W(MLOGP) was −1.71, number of rotatable bonds was 2, TPSA was 58.92 A2, log S (ESOL) was −3.82, GI absorption was high, BBB permeant was “yes,” log Kp (skin permeation) was −6.54 cm/s, and bioavailability score was 0.55 (Table 2). The properties of phellodendrine were in accordance with the RO5, implying that it had drug potential.

Table 2 Molecular properties of phellodendrine.

Property	Value	
Molecular weight	342.41 g/mol	
nHAcc	4	
nHDon	2	
LogP0/W (MLOGP)	−1.71	
Rotatable bonds	2	
TPSA	58.92A2	
Log S (ESOL)	−3.82	
Molar refractivity	100.92	
Log Kp (skin permeation)	−6.54 cm/s	
Bioavailability score	0.55	

Common targets between phellodendrine-related targets and inflammation-related targets

Figure 2 shows 98 potential target genes of phellodendrine and 837 inflammation-related targets. The 12 common targets between phellodendrine-related targets and inflammation-related targets were determined through a Venn diagram. The 12 common targets were PTGS1, PTGS2, ADRB2, F3, HTR1A, DPP4, ESR1, ABCC1, ABCB1, JUN, ABCG2, and PIK3CA (Table 3).

Figure 2 Targets of phellodendrine against inflammation.

(A) Targets of phellodendrine. (B) Targets of inflammation. The blue circle represents the targets from the GeneCards databases, the yellow circle represents the targets from the Drugbank databases, and the green circle represents the targets from the OMIM databases. (C) Classification of phellodendrine target genes for the top 50 according to the biochemical criteria. (D) Twelve overlapping targets between phellodendrine and inflammation. The blue circle represents the phellodendrine targets, and the yellow circle represents the inflammation target.

Table 3 Common targets between phellodendrine-related targets and inflammation-related targets.

Uniprot ID	Target	Description	
P23219	PTGS1	Cyclooxygenase-1, Prostaglandin G/H synthase 1	
P35354	PTGS2	Cyclooxygenase-2,Prostaglandin G/H synthase 2	
P07550	ADRB2	Beta-2 adrenergic receptor 2	
P13726	F3	Coagulation factor VII/tissue factor	
P08908	HTR1A	Serotonin 1a (5-HT1a) receptor	
P27487	DPP4	Dipeptidyl peptidase IV	
P03372	ESR1	Estrogen receptor alpha	
P33527	ABCC1	Multidrug resistance-associated protein 1	
P08183	ABCB1	P-glycoprotein 1	
P05412	JUN	Proto-oncogene c-JUN	
Q9UNQ0	ABCG2	ATP-binding cassette sub-family G member 2	
P42336	PIK3CA	PI3-kinase p110-alpha subunit	

Network of common targets

The 12 common targets were imported to the STRING database for analysis and PPI network establishment. Figure 3A shows 12 nodes, 18 edges, and an average node degree of 3. The “Analysis network” tool in Cytoscape software was used to analyze the topological features of common targets. PTGS2 (degree = 6), ABCB1 (degree = 5), JUN (degree = 5), ESR1 (degree = 5), ABCG2 (degree = 3), ABCC1 (degree = 3), PIK3CA (degree = 3), F3 (degree = 3), PTGS1 (degree = 2), and DPP4 (degree = 1) were the important targets in the network (Fig. 3B).

Figure 3 Network of common targets.

(A) PPI network of phellodendrine for the treatment of inflammation with the STRING database. (B) The optimized PPI network constructed using Cytoscape. Node size and color from olive green (high) to light green (low) represent the degree.

GO biological process included biological process (BP), cellular component (CC), and molecular function (MF) to explain anti-inflammation biological processes at common targets. As shown in Fig. 4A, the results of GO enrichment analysis displayed 21 BPs, three CCs, and eight MFs (P < 0.05). The top 10 significantly enriched BP terms were response to drug, negative regulation of smooth muscle contraction, xenobiotic transport, cyclooxygenase pathway, angiogenesis, prostaglandin biosynthetic process, drug transmembrane transport, behavioral fear response, endothelial cell migration, and brown fat cell differentiation. The three CC terms were plasma membrane, apical plasma membrane, and cell surface. The eight MF terms were ATPase activity, coupled to transmembrane movement of substances, protein homodimerization activity, prostaglandin-endoperoxide synthase activity, xenobiotic-transporting ATPase activity, transporter activity, peroxidase activity, enzyme binding, and protein binding.

Figure 4 GO and KEGG enrichment analyses.

(A) GO analysis of common targets. Top 10 significantly enriched terms in BP. GO enrichment analysis showed three and eight remarkably enriched items in CC and MF, respectively. (B) Bubble chart of the significantly enriched terms in KEGG pathways.

The six signaling pathways of KEGG enrichment comprised regulation of lipolysis in adipocytes, ABC transporters, cAMP signaling pathway, estrogen signaling pathway, TNF signaling pathway, and serotonergic synapse (Fig. 4B).

Protective effects of phellodendrine

To detect the suitable concentration of phellodendrine on RAW 264.7 cells, we treated RAW 264.7 cells with phellodendrine at concentrations of 5, 10, 20, 40, 80, and 160 µg/mL for 24, 48, and 72 h. Figure 5A shows no significant difference in cell viability between the 5 µg/mL group and the control group for 24, 48, and 72 h. Other concentrations demonstrated inhibitory effects on cells at different times.

Figure 5 Effect of phellodendrine on the viability of RAW264.7.

(A) Cell viability of RAW264.7 treated with phellodendrine (5, 10, 20, 40, 80, and 160 mg/L) for 24, 48, and 72 h. (B) Cell viability of RAW264.7 incubated with LPS and then treated with phellodendrine (0, 5, 10, 20, 40, 80, and 160 mg/L) for 24, 48, and 72 h. Data are the mean ± SEM. *P < 0.05, **P < 0.01 vs. the control group (0); #P < 0.05, ##P < 0.01, vs. the LPS group.

RAW 264.7 cells were administered with LPS, an approved inflammation model. Phellodendrine at 5, 10, 20, and 40 µg/mL significantly increased the viability of cells treated with LPS at different times (P < 0.01 or 0.05 Fig. 5B). These results demonstrated the protective effects of phellodendrine on RAW 264.7 cells treated with LPS. Therefore, three concentrations of 5, 10, and 20 µg/mL were used to act on the cells in subsequent experiments.

Effects of phellodendrine on the levels of IL-6 and IL-1β

The levels of IL-6 and IL-1β increased significantly after LPS administration for 24 or 48 h than those of the control group (P = 0.011, P = 0.031, P = 0.032). Compared with the LPS group, phellodendrine reduced the levels of IL-6 only in 24 h (P = 0.014) and slightly decreased IL-1β levels (Figs. 6A and 6B).

Figure 6 Effects of phellodendrine on IL-6 and IL-1β levels in RAW264.7.

Various concentrations of phellodendrine (5, 10, 20, 40, 80, and 160 mg/L) with LPS or only LPS were added to cells for 24, 48, and 72 h. (A–C) IL-6 levels. (B–D) IL-1β levels. Data are the mean ± SEM. *P < 0.05, **P < 0.01 vs. the control group (0); #P < 0.05, ##P < 0.01, vs. the LPS group.

At 72 h, IL-6 declined in the LPS group compared with that in the control group (P = 0.043). IL-1β increased in the LPS group compared with that in the control group (P = 0.007) and declined in the phellodendrine group (except 5 and 10 µg/mL) compared with that in the LPS group (P = 0.018, P = 0.018, P = 0.03, P = 0.005; Figs. 6C and 6D).

Putative target validation

After different interventions, the expression levels of PTGS1 mRNA, HSP90ab1 mRNA, and AKT1 mRNA in the LPS group were significantly lower than those in the control group (P = 0.000, P = 0.036, P = 0.000) and phellodendrine group (P = 0.010, P = 0.016, P = 0.005, 20 mg/L) at 24 h. Compared with the control group, LPS could decrease the expression of HTR1A mRNA, PI3CA mRNA, and F10 mRNA (P = 0.004, P = 0.001, P = 0.003). By contrast, phellodendrine could increase the expression of HTR1A mRNA and PI3CA mRNA (P = 0.000, P = 0.008, P = 0.020) compared with the LPS group at 48 h (Fig. 7).

Figure 7 Effects of phellodendrine on mRNA expression of targets in RAW264.7.

Various concentrations of phellodendrine (5, 10, and 20 mg/L) with LPS or only LPS were added to cells for 24, 48, and 72 h. Relative mRNA expression of PTGS1, PTGS2, HSP90ab1, AKT1,HTR1A, PI3CA, ADRB2, and F10. L represents LPS. P represents phellodendrine. Data are the mean ± SEM. *P < 0.05, **P < 0.01 vs. the control group (0); #P < 0.05, ##P < 0.01, vs. the LPS group.

Unexpectedly, LPS administration for 72 h significantly increased the expression of HTR1A mRNA, PI3CA mRNA, and F10 mRNA (P = 0.000, P = 0.010, P = 0.000) but decreased their expression in the phellodendrine group (P = 0.000, P = 0.000, P = 0.000; P = 0.033; P = 0.032, P = 0.000 P = 0.000). The expression of PTGS2 mRNA increased in the LPS group and decreased in the phellodendrine group (P = 0.003, P = 0.000, P = 0.000).

Discussion

Well-known herbal formulas such as Ermiao Powder, Sihuang Powder, Huanglian Shangqing Pill, DaBuYinWan, and Bushen Mingmu Pill containing Cortex phellodendri have been used clinically to treat inflammation-related diseases in China. However, the mechanism of Cortex phellodendri on inflammation is unclear. The underlying mechanism of phellodendrine (one of the Q-markers of Phellodendri Chinensis Cortex) on inflammation should be investigated for clinical use.

In this study, we found that phellodendrine had drug potential and identified 98 potential target genes, 837 inflammation-related targets, and 12 common targets. Additionally, pathway and functional enrichment analyses revealed that phellodendrine regulated the cAMP, estrogen, TNF, and serotonergic synapse signaling pathways. The anti-inflammation effect of phellodendrine was verified in RAW 264.7 cells exposed to LPS. The levels of IL-6 decreased after treatment with phellodendrine. Moreover, phellodendrine changed the mRNA expression of eight valuable putative targets: PTGS1, PTGS2, HSP90ab1, AKT1, HTR1A, PI3CA, F10, and ADRB2.

IL-6 is a pleiotropic cytokine (including pro-inflammatory cytokine) that plays an important role in host defense by modulating immune and inflammatory responses. Anti-inflammation has recently attracted attention as a strategy for the treatment of many diseases. IL-6 plays an important role in the inflammatory response with diseases. A linear relationship was found between ischemic stroke risk and circulating IL-6 (potential target for reducing the risk of ischemic stroke) levels (Papadopoulos et al., 2021). A causal role of IL-6 and soluble IL-6 receptor in vascular disease or depression was also reported recently (Kelly, Smith & Mezuk, 2021; Swerdlow, Holmes & Kuchenbaecker, 2012; Georgakis et al., 2020; Georgakis et al., 2021). Therefore, IL-6 signaling has been proven as one of the most hopeful targets for anti-inflammation in the treatment of diseases. Our study also showed that phellodendrine significantly reduces IL-6 levels induced by LPS. According to previous studies, phellodendrine may act as a drug targeting IL-6, and it can treat or prevent inflammation-related diseases.

Valuable putative genes that are differentially expressed are related to inflammation treated with phellodendrine, thereby suggesting their various roles in the course of inflammation. Only PTGS2 expression increased in the LPS group but decreased in the phellodendrine group. Prostaglandin-endoperoxide synthase (PTGS1,2) or cyclooxygenase (COX-1,-2) are crucial enzymes in the prostaglandin synthesis pathway, which is critical for the regulation of inflammatory processes (Coghill et al., 2011). PTGS1 and PTGS2 play similar roles in the prostaglandin synthesis pathway, but they differ in expression pattern. PTGS1 is constitutively expressed in prostanoid-producing cells in most tissues, whereas inducible expression of PTGS2 is triggered by cytokines and hypoxia (Smith, Garavito & DeWitt, 1996). Our experimental data revealed that PTGS2 mRNA concentrations significantly increased in the inflammatory response stimulated by LPS, and anti-inflammatory phellodendrine could decrease the expression. These results were consistent with the previous study that non-steroidal anti-inflammatory drugs interact with the prostaglandin synthesis pathway aimed to inhibit inflammation, whereas quercetin and kaempferol decreases the level of PTGS2 mRNA in Chang liver cells, showing anti-inflammatory effects (García-Mediavilla et al., 2007).

Intimate cooperation and interaction between coagulation and inflammation is well known (Cirino et al., 2000; Graf et al., 2019; Doronin et al., 2012; Levi, van der Poll & Büller, 2004). As shown in many vitro studies, coagulation proteases play an important role in increasing proinflammatory cytokine expression. Coagulation factor X (FX, active form is FXa) encoded by the F10 gene is a vitamin K-dependent serine protease that has a crucial function in blood coagulation. FXa stimulates the production of IL-6, IL-1, and IL-8 (Senden et al., 1998; van der Poll, de Jonge & Levi, 2001). In line with these findings, F10 mRNA increased in the LPS group but decreased after treatment with phellodendrine for 72 h; the IL-1β levels showed similar results in our study. F10 mRNA decreased after LPS administration for 24 or 48 h, whereas IL-6 and IL-1β increased. The reasons for the different results may be as follows. First, we detected the mRNA level of F10, but protein levels were affected. The possible reason for the decrease in mRNA level is that more mRNA is used to synthesize protein. Second, the administration time may vary. The content of mRNA levels may increase in the future, not at 24 or 48 h. Third, inflammatory cytokines bidirectionally interacted with the coagulation system. For example, IL-6 is involved in the initiation of the coagulation system. Therefore, a certain time must pass to start the coagulation cascade after the inflammatory cytokines were elevated.

HTR1A and PIK3CA were enriched in the cAMP signaling pathway. HTR1A is the gene of 5-hydroxytryptamine receptor 1A (5-HT1A), which is a G-protein coupled receptor for 5-hydroxytryptamine. 5-HT1A (HTR1A) is closely associated with the pathogenesis of inflammation-related diseases, such as depression and anxiety, in humans and other animals (Xia et al., 2019; Staes et al., 2019; Wang, Wang & Chen, 2021). Zhike-Houpu Herbal Pair increased the protein and mRNA expression of HTR1A, thereby relieving depressive behaviors (Xia et al., 2019). Additionally, microRNA-26a-2 plays an antidepressant role by targeting HTR1A(Xie et al., 2019). Activated HTR1A and HTR4 by serotonin exert anti-inflammatory effects and inhibit HTR1A, and HTR4 plays the opposite effect (Wang et al., 2020). These findings were in line with our results that phellodendrine increased the expression of HTR1A mRNA in RAW264.7 subjected to LPS at 48 h. 5-HT1A coupled with G protein inhibited the activity of downstream effector adenylate cyclase and decreased the production of cAMP. cAMP regulated the PI3K/Akt pathway, and activated PKA regulated the NF-κB pathway. PIK3CA is an important part of the previously mentioned PI3K/Akt pathway. PIK3CA (encoding p110α, the catalytic subunit of PI3K) oncogenic mutations are involved in various types of cancer, such as colorectal cancer, breast cancer, ovarian cancer, hepatocellular carcinoma, and hidradenoma papilliferum (Arafeh & Samuels, 2019; Goto et al., 2017). After the activation of the PI3K/Akt pathway, activated Akt leads to the activation of NF-κB. Subsequently, NF-κB can induce the expression of pro-inflammatory factors (such as IL-6 and IL-1β) and some enzymes involved in the inflammatory cascade (such as iNOS and PTGS2). Our results showed that phellodendrine reduced the PI3CA mRNA level raised significantly by LPS in RAW264.7 for 72 h and altered AKT1 expression, which may inhibit the effect of NF-κB through the inhibition of the PI3K/Akt pathway, and finally alleviate the inflammatory response. These findings were in line with our results that the cAMP signaling pathway was involved in the anti-inflammatory effect of phellodendrine according to network pharmacology analysis.

Conclusions

In summary, network pharmacological analyses and experimental validation were conducted to uncover the pharmacological mechanism of phellodendrine on inflammation. In this study, 12 common targets against inflammation were acquired through network pharmacology. Phellodendrine exerted anti-inflammation effects through the cAMP signaling pathway, TNF signaling pathway, serotonergic synapse, and so on. Further validation with experimental evidence demonstrated that phellodendrine inhibited inflammation in RAW264.7 cells by regulating the expression of IL-6, IL-1β, PTGS1, AKT1, HSP90ab1, HTR1A, and PI3CA. Our research might provide new treatment regimens for inflammation-related diseases. However, details about the anti-inflammation of phellodendrine need to be further investigated.

Supplemental Information

Supplemental Information 1 Common targets.

Click here for additional data file.

Supplemental Information 2 Common targets between phellodendrine-related targets and inflammation-related targets.

Click here for additional data file.

Supplemental Information 3 Targets of phellodendrine.

Click here for additional data file.

Supplemental Information 4 The relationship between targets and phellodendrine.

Click here for additional data file.

Supplemental Information 5 Raw GO and KEGG data.

Click here for additional data file.

Supplemental Information 6 RT-PCR.

Click here for additional data file.

Supplemental Information 7 Protective effects of phellodendrine.

Click here for additional data file.

Supplemental Information 8 Effects of phellodendrine on the levels of IL-6 and IL-1β.

Click here for additional data file.

Abbreviations

AKT serine/threonine kinase

AlogP partition coefficient between octanol and water

GO gene ontology

Hacc hydrogen acceptor count

Hdon hydrogen donor count

IL-1 interleukin 1

IL6 interleukin 6

KEGG Kyoto Encyclopedia of Genes and Genomes

PI3K phosphoinositide 3-kinase

PPI protein–protein interaction

STRING Search Tool for the Retrieval of Interacting Genes/Proteins

TCMSP Traditional Chinese Medicine Systems Pharmacology Database and Analysis Platform

TNF tumor necrosis factor

Additional Information and Declarations

Competing Interests

Author Contributions

Data Availability

The authors declare that they have no competing interests.

Lili Hu conceived and designed the experiments, performed the experiments, prepared figures and/or tables, authored or reviewed drafts of the article, and approved the final draft.

Jue Wang performed the experiments, authored or reviewed drafts of the article, and approved the final draft.

Na Wu performed the experiments, analyzed the data, prepared figures and/or tables, and approved the final draft.

Xiaoge Zhao analyzed the data, prepared figures and/or tables, and approved the final draft.

Donghui Cai performed the experiments, authored or reviewed drafts of the article, and approved the final draft.

The following information was supplied regarding data availability:

The raw measurements are available in the Supplemental Files.

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
