# Peer review of "Utilizing network pharmacology and experimental validation to investigate the underlying mechanism of phellodendrine on inflammation"

_PeerJ, doi:10.7717/peerj.13852_

## Round 0.1 · original submission · Major Revisions

Two reviewers have given their comments and please be advised the paper requires major revisions. Please incorporate the revisions and resubmit at the earliest.

·

Basic reporting

Computational investigations supporting with experimental validation are profoundly important. In the present article with title “Utilizing network pharmacology and experimental validation to investigate the underlying mechanism of phellodendrine on inflammation” authors attempted finding mechanism of phellodendrine in the inflammation.
The positive strengths of the present manuscript are good concept, elaborative results, graphical and tabular analysis.
However, the study needs extensive revision before acceptance;
1. Manuscript need to rewrite proper English and without any grammatical and typographical mistakes.
2. Line 51 -Write biological name as per the nomenclature-
3. Rewrite the rational of drug selection for proposed study with relevant references.
4. Whether the Cortex phellodendri is the only source of phellodendrine? Clarify..
5. Rewrite the Introduction with reference to the objective of the study
6. Flow in discussion section found missing.

Experimental design

1. Kindly provide references for the experimental section2.5.2 to 2.5.5.
2. Please mention how network was analyzed meaning how it was treated and which variable was considered for the constructed network analysis.
3. Authors need to rewrite experimental part 2.2 for better understanding.

Validity of the findings

No Comment

Additional comments

No Comments

Reviewer 2 ·

Basic reporting

Professional English is been used. In supplementary files the data set names were given in Chinese which is unclear.

Literature part is good.


Raw data shared is informative.

Results were good.

Experimental design

No comments

Validity of the findings

Usage of SPSS software for statistical analysis should be justified. Why not Graph pad prism is used?

Additional comments

Justify why SPSS software is been used.

---

## Round 0.2 · accepted · Accept

Based on reviewers assessment I am happy to consider paper for publication.

Reviewer 2 ·

Basic reporting

Everything good

Experimental design

Arranged as per reviewer's comments. It's good.

Validity of the findings

Results reported were meeting the objective. No comments in this regard.

Additional comments

No comments